# Exploring Household Food Waste Reduction for Carbon Footprint Mitigation: A Case Study in Shanghai, China

**DOI:** 10.3390/foods12173211

**Published:** 2023-08-25

**Authors:** Chang Liu, Jie Shang, Chen Liu

**Affiliations:** 1School of Economics and Management, Northeast Forestry University, Harbin 150040, China; chang1127@nefu.edu.cn; 2Sustainable Consumption and Production Area, Institute for Global Environmental Strategies, 2108-11 Kamiyamaguchi, Hayama 240-0115, Japan

**Keywords:** edible food waste, carbon footprint, life cycle, influencing factors

## Abstract

With the steady growth of the global population and the accelerated urbanization process, the carbon footprint resulting from food waste has a significant impact on the environment and sustainable development. Considering Shanghai’s significance as a major urban center in China and a global hub for economic and cultural activities, this study primarily aims to accurately estimate household food waste generation and calculate the carbon footprint related to edible food waste. It analyzes the factors influencing household food waste generation and reviews the anti-food waste-related policies at both the national and Shanghai regional levels. The study reveals that although the Shanghai municipal government attaches great importance to the issue of food waste, the current policies mainly focus on the catering industry, and there is still a need for further strengthening measures to address food waste at the household level. In Shanghai, the per capita daily food waste generation is 0.57 kg, with 43.42% being edible food waste, contributing to a per capita daily carbon footprint of 1.17 kgCO_2_eq. Employing the logistic regression analysis to scrutinize the characteristics of the respondents, it is ascertained that education level and annual household income significantly influence food waste generation. In addition, excessive food quantities and expiration dates lead to high-frequency food waste. The culmination of this study is the formulation of a series of pragmatic and impactful policy recommendations aimed at curbing the carbon footprint that stems from food waste.

## 1. Introduction

Global greenhouse gas (GHG) emissions have been consistently increasing over the past few decades. According to data from the International Energy Agency (IEA), global energy-related carbon dioxide (CO_2_) emissions increased by 321 Mt in 2022, reaching a new record high of over 36.8 Gt [1]. China is one of the major emitters of greenhouse gases, and in 2022, China’s carbon dioxide (CO_2_) emissions reached 11.48 billion tonnes, accounting for 31.19% of the global emissions [1,2]. Among the numerous sources of carbon emissions, the issue of carbon emissions resulting from food waste is often overlooked. However, the impact of food waste on the environment is increasingly severe. According to the United Nations Environment Programme’s “2021 Food Waste Index”, greenhouse gas emissions from food waste account for 8% to 10% of global greenhouse gas emissions [3]. It is estimated that globally, around 1.3 billion tons of food is wasted each year, which is equivalent to one-third of the total global food production [4], causing a loss of 936 billion dollars [5]. According to UN data, about 14% of total global food production is lost between harvest and retail. Another 17% is wasted (11% in households, 5% in food service, and 2% in snacks) [6]. In China, the issue of food waste is particularly severe, with approximately 27% (3.49 ± 4 Mt) of annually produced food intended for human consumption being lost or wasted [7], exacerbating the contradiction between production inputs and environmental pressures [8].

There are four main aspects of research on food waste carbon emissions and carbon footprint, namely the carbon emissions and carbon footprint of the catering industry [8,9,10], the carbon emissions and carbon footprint of universities [11,12,13], the carbon emissions and carbon footprint of households [14,15,16], and the impact of food waste management and utilization on carbon emissions and carbon footprint [17,18]. The restaurant industry faces significant challenges in terms of food waste and carbon footprint. A study estimated the total carbon footprint generated by food waste in the catering sector in Beijing. The results showed that the total carbon footprint of food waste ranged from 1,925,100 to 2,085,200 metric tons of CO_2_eq. Among these, the largest carbon emissions were from the agricultural production stage, followed by the consumption stage, and then the food waste disposal stage [8]. An estimation study of the carbon footprint of food waste in Malaysia’s casual restaurants revealed that the largest food losses occurred during preparation (51.37%), followed by service losses (30.95%) and plate waste (17.8%). Meanwhile, the total average electricity consumption and carbon footprint in the selected three regions were found to be 197.67 kWh and 19.63 kgCO_2_eq [9]. A study conducted a life cycle assessment to calculate the environmental impact of food waste in restaurants. The results showed that the carbon footprint generated by each wasted plate of food ranged from 128 gCO_2_eq to 324 gCO_2_eq. The main wasted products were rice and legumes, followed by beef, and then other carbohydrates [10]. As places with a large population gathering, university campuses and student food consumption in cafeterias have a certain impact on food waste and carbon footprint. A field survey conducted on 9660 questionnaires from 30 universities in China revealed that the total annual food waste generated by students nationwide ranges from 1.3362 million to 1.3773 million metric tons. The carbon emissions of a pork ribs meal were estimated to be 1.6758 kgCO_2_eq. Among the various stages, the manufacturing of raw materials had the highest carbon footprint, accounting for 82% of the total, followed by food preparation and raw material manufacturing with 11% and 6%, respectively. The packaging and disposal stage had the lowest carbon footprint, accounting for only 1% [11]. A survey was conducted at a university in Portugal with approximately 7000 students, and revealed a substantial food waste rate of around 13.4%. The estimated monthly economic loss due to food waste was EUR 3080, and the ecological footprint was 2.8 gha. The total amount of food waste was 417 kg [12]. A survey conducted with 9192 samples from 29 provinces in China estimated that university cafeterias wasted 1.55 million tons of food in 2018, resulting in a related carbon footprint of 2.51 million metric tons of CO_2_eq. The two food categories that contributed the most to the total carbon footprint were meat, accounting for 46.28%, and grains, accounting for 36.52% [13]. Households are one of the most significant sources of food waste, and household food waste has a significant impact on carbon footprint. A survey estimated the greenhouse gas emissions from food waste in Canadian households and found that municipal authorities play an important role in incentivizing and facilitating behaviors to reduce household food waste and associated greenhouse gas emissions [14]. A study evaluating the carbon emissions from household food waste in South Korea found that food waste contributed approximately 0.73 ± 0.06 kg (per household per day) and 0.71 ± 0.05 kgCO_2_eq greenhouse gas emissions. Furthermore, the study revealed that animal-based food waste had relatively higher environmental and economic losses compared to non-animal-based food waste [15]. A survey quantified the carbon footprint of 17,110 household members in China, covering 1935 types of food. It was found that, on average, each person wastes (consumed) 16 (415) kg of food at home per year, which is equivalent to a carbon footprint of 40 (1080) kg of carbon dioxide. Additionally, vegetables, rice, and wheat were the most consumed and wasted food items. Apart from these three plant-based foods, pork and seafood also made significant contributions to the embedded footprint [16]. The handling and utilization of food waste also have a significant impact on carbon footprint, and modern technological methods contribute to addressing the carbon emissions associated with food waste management. A study investigated the impact of food waste transfer on the carbon footprint using Canadian composting data, finding that composting of food waste reduced the total carbon footprint by 138 million tonnes and 1.33 million tonnes of carbon dioxide, accounting for approximately 18% and 20% of the total carbon footprint of urban solid waste in Canada, respectively [17]. A literature review conducted a comprehensive analysis of food waste utilization technologies with low carbon footprints. Technologies such as hydrothermal carbonization (HTC), dendro liquid energy (DLE), and ultra-fast hydrolysis can all reduce the carbon footprint of food waste [18].

In recent times, an evolving awareness has emerged, delineating the ramifications of food waste, extending beyond financial detriments and resource squandering to encompass pronounced ecological ramifications [11,13,15]. Consequently, a considerable number of researchers have embarked on comprehensive studies of the carbon footprint generated by food waste [18,19,20]. Household food waste occupies a significant position in the entire food supply chain and makes a substantial contribution to the total amount of food waste [21,22,23]. Every household generates a certain degree of food waste, making it a widespread issue overall. Studying the carbon footprint of household food waste allows us to assess the environmental impact of household food waste and provides valuable information for developing strategies and policies to reduce carbon emissions. Shanghai, as an internationally influential metropolis, is driving the development of China. However, there has been a lack of in-depth research conducted by scholars on the carbon footprint of household food waste. In addition to that, previous studies have often categorized food waste by specific food items, while this study simplifies the classification into five categories: cooking waste, leftover of cooked staple food, leftover of cooked dishes, untouched food, and tea leaves and coffee grounds. This simplified classification method is more concise and suitable for analyzing and managing overall food waste issues. Compared to categorizing by specific food items, this classification method divides food waste into several common categories, covering different stages and scenarios of waste generation. It is also easier to understand and apply, making it suitable for a wide range of situations. Studying the carbon footprint of household food waste in Shanghai has several implications: (1) Estimating specific data: Research can provide specific data on food waste in Shanghai, including the amount of waste, carbon emissions, and carbon footprint. These data can serve as a scientific basis for developing effective strategies and measures to reduce waste and carbon emissions for governments, organizations, and individuals. (2) Understanding waste patterns: Research can help identify and understand the main causes and patterns of food waste in Shanghai. By analyzing the types and extent of waste, effective measures can be developed to reduce food waste, optimize resource utilization, and improve the sustainability of the food supply chain. (3) Environmental awareness: Food waste contributes to resource waste, energy consumption, and greenhouse gas emissions, leading to climate change and environmental pollution. Studying the specific environmental impact of food waste in Shanghai can deepen people’s understanding of its environmental damage and promote environmental awareness. (4) Policy implications: Research can provide a theoretical basis for developing policies and measures to combat food waste and reduce carbon emissions in Shanghai. It can inform the formulation of policies to reduce waste and promote sustainable consumption. Moreover, the findings can serve as a reference for similar policies in other regions of China. Therefore, this article takes Shanghai as an example and first determines the system boundaries of food waste carbon footprint. Then, based on survey data on household food waste in Shanghai, the carbon footprint of household food waste in Shanghai is calculated, and its influencing factors are analyzed. Furthermore, a comprehensive review of anti-food waste policies in China and Shanghai is conducted. Finally, effective policy recommendations are proposed based on the research findings, with the aim of reducing food waste.

In summary, studying the carbon footprint of household food waste in Shanghai can provide specific data, identify waste patterns, reveal environmental impacts, raise awareness of environmental damage, and provide a theoretical foundation for developing policies for anti-food waste and carbon reduction. Additionally, it can serve as a reference for related policies in other regions of China.

## 2. Materials and Methods

### 2.1. Investigation of Food Waste Generation

#### 2.1.1. Sampling Size and Analytical Methodology

During the period from December 2022 to February 2023, a comprehensive telephone and online questionnaire survey was conducted among Shanghai residents, providing a snapshot of their food waste situation. The research was conducted by a professional research firm, Cedian Consulting Co. (Beijing, China), which has extensive experience and a professional team in the field of market research. This investigation utilized stratified random sampling, with geographic regions serving as the basis for creating sampling strata. Sampling was performed considering the population size of each region to attain more precise and dependable approximations of the entire population. Cedian Consulting Co. randomly selected citizens aged 18 and above according to the population proportion of each region. Additionally, during the sampling process, factors such as gender, age, educational level, and family annual income were taken into consideration. Specifically, the survey utilized a stratified random sampling procedure. Firstly, the population of Shanghai was stratified based on geographical regions, and 16 regions were chosen as sampling units. Subsequently, the determination of the sample size for each region was guided by its population size to guarantee the comprehensive representation of each region’s characteristics. Lastly, random sampling was conducted within each sampling unit to ensure equal opportunities for all units to be included in the sample, thereby ensuring the representativeness and reliability of the sample [24]. The results of the seventh population census show that the resident population of Shanghai is 24.87 million [25]. In this study, we had a sample size of 461 individuals, surpassing the minimum sample size calculated through the formula [26]. The following is the equation used to determine the minimum required sample size.
(1)$$n=\frac{P\cdot \left(1-P\right)}{\frac{E^{2}}{Z^{2}}+\frac{P\cdot \left(1-P\right)}{N}}$$

*P* denotes the probability value, *P* = 0.5. *E* signifies the margin of error, set at *E* = 0.05 for a 5% confidence interval. *Z* represents the confidence coefficient, where *Z* = 1.96 for a 95% confidence level. *N* represents the total population, while *n* signifies the minimum required sample size. Thus, through computation, the minimum sample size for a 95% confidence level and a 5% confidence interval is determined as 384. Our chosen sample size of 461 surpasses this minimum requirement, affirming its representative nature.

The public questionnaire survey had a sample size of 461 respondents. Among them, 280 samples were collected from the eight main urban areas, and 181 samples were collected from the surrounding areas. The specific distribution of the samples in each region is shown in Figure 1.

#### 2.1.2. Content of Questionnaire

Conducting a combination of telephone and online surveys can broaden the scope of the research, improve response rates, diversify data collection methods, and enable data validation and comparison, thus obtaining more comprehensive and reliable research results. This questionnaire is primarily composed of four sections. The initial segment is designed to collect fundamental information from participants, encompassing factors like gender, age, occupation, educational background, and annual household income. This is achieved through the use of closed-ended and factual questions. The second segment delves into the present state of food waste production and handling, encompassing metrics like the volume of food waste generated and the strategies employed for disposal. This section of the questionnaire utilizes quantitative questions and semi-open-ended questions. The third segment delves into the principal factors contributing to food waste generation among participants. This section employs closed-ended questions, including conditional logic questions and multiple-choice questions. Lastly, the fourth segment probes participants’ viewpoints regarding the mitigation of food waste. This section utilizes a rating scale approach, employing the Likert scale.

Household food waste behavior is a complex and multifaceted phenomenon, influenced by a combination of various factors. In the process of designing the survey questionnaire, we thoroughly integrated insights from the literature review, practical observations, and accumulated experiences. Based on these backgrounds, we have developed a preliminary hypothesis that factors such as region, gender, age, educational level, and annual household income may be associated with household food waste behavior. We conjecture that these factors might exert a certain influence on household food waste behavior, leading to varying degrees of food wastage. Specifically, we speculate that certain regions might be more prone to higher levels of food waste [27,28,29], specific gender and age groups could exhibit a greater tendency for food waste generation [28,29,30], while educational level and annual household income might be linked to the frequency and quantity of food waste practices [29,31,32]. The design of questions regarding the influencing factors of food waste is based on multiple sources, including the literature review, practical observations, and accumulated experience. Building upon these foundations, we have drawn insights from similar existing studies and relevant questionnaire design methods. The aim is to ensure that the survey questions encompass a wide range of potential factors that could influence food waste. This comprehensive approach is intended to provide a thorough understanding of household food waste behavior and its various associated influencing factors.

### 2.2. Carbon Footprint Measurement

#### 2.2.1. System Boundaries

This article estimates the carbon footprint of food waste using a life cycle assessment, which is a method for evaluating the greenhouse gas emissions of a product, service, or activity throughout its entire life cycle. Unlike traditional carbon footprint calculations, life cycle assessment takes into account greenhouse gas emissions from various stages such as raw material extraction, manufacturing, distribution, use, and disposal [33,34]. This article divides the life cycle of food waste into four stages: the raw material production stage, the transportation stage, the food preparation stage, and the waste disposal stage. The carbon footprint of food waste throughout its life cycle is calculated step by step based on the four stages mentioned above. The specific system boundary is set as shown in Figure 2.

#### 2.2.2. Carbon Footprint Calculation Methods and Data Sources

Carbon footprint is typically expressed in terms of carbon dioxide equivalent (CO_2_e), which is a unit that converts the emissions of various greenhouse gases into the equivalent amount of carbon dioxide [35,36]. This conversion allows for a standardized and comparable representation of the overall impact of different greenhouse gases on climate change. By expressing the emissions in CO_2_e, it becomes easier to understand and compare the carbon footprints of different activities, products, or processes [37]. The formula for calculating carbon emissions at each stage of food waste is as follows:1.Raw material production stage:
Carbon Emissions = Amount of food waste × Carbon emissions coefficient of food production

2.Transportation stage:

Carbon Emissions = Amount of food waste × Carbon emissions coefficient of transportation

3.Food preparation stage:

Carbon Emissions = Energy Consumption × Carbon emission coefficient of fuel usage

4.Waste disposal stage:

Carbon Emissions = Amount of food waste × Carbon emission factor of incineration

When calculating the carbon footprint of food waste, it is essential to conduct surveys on the various components of the food waste lifecycle to gather data on activities at each stage. By accumulating and summing up this data, the calculation of the carbon footprint can be obtained [11,38]. The specific calculation formulas are as follows:(2)$$CF=\underset{i=1}{\sum }Q_{i}\times EF_{i}$$

*CF* represents the carbon footprint, *Q_i_* represents the quantity or intensity data of the substance or activity (mass/volume/kilometers/kilowatt-hours), and *EF_i_* represents the unit carbon emission factor (CO_2_eq/unit). The carbon emission coefficients for each stage are as follows:1.Raw material production stage

In leftover cooked staple food, the difference in waste quantity between leftover noodles and rice is not significant; therefore, their carbon emission factor is the average of rice and wheat [39,40]. In leftover cooked dishes, the waste quantity of plant-based foods is twice that of animal-based foods [8]; thus, its emission factor is the average of twice the plant-based foods and animal-based foods. Unprocessed food refers to expired food, including plant-based foods, animal-based foods, tea, coffee, and so on. Its emission factor is the average of these three categories. As for tea and coffee, their carbon emission factor is the average of the two. As shown in Table 1, the specifics are as follows [38,39,40,41,42,43,44].

2.Transportation stage

The food in Shanghai mainly comes from Chongming District and the surrounding areas, with transportation primarily relying on road transport. The average distance from these areas to Shanghai is approximately 60 km. Therefore, in this study, we have set the transportation distance as 60 km for road transport, assuming the use of 3.5t trucks at full capacity. For specific carbon emission factors, please refer to Table 2 [11,38].

3.Food preparation stage

In the food preparation stage, the main consumption is related to water, electricity, natural gas, and other resources. The corresponding carbon emission factors can be found in Table 3 [11,38].

For cooking grains, we will assume a quantity of 500 g, with 1000 g of water, using a rice cooker with a power rating of 900 W, and a cooking time of 35 min. For vegetables, we will assume a quantity of 500 g cooked for 2 min. For meat, we will assume a quantity of 500 g cooked for 40 min. Since the majority of residents in Shanghai use natural gas for cooking, we will use natural gas for calculations. The average consumption of natural gas is approximately 0.4 m^3^/h [45]. During the food preparation stage, processing is required only for cooking, vegetable preparation, coffee, and tea. The specific carbon emission factors can be found in Table 4.

4.Waste disposal stage

During the food waste disposal stage, following the implementation of waste classification in Shanghai, food waste is categorized as wet waste. The primary methods of its treatment encompass composting and anaerobic digestion [46,47,48]. The relevant carbon emission factors can be found in Table 5 [49].

### 2.3. Cronbach’s α Reliability Coefficient Analysis

Cronbach’s α is a measure of the internal consistency or reliability of a scale or test. In this study, we utilized Cronbach’s α to analyze the reliability of the survey questionnaire. Cronbach’s α values range from 0 to 1. A value closer to 1 indicates stronger internal consistency, meaning that the items in the scale are closely related and the measurement of the same underlying construct is more reliable [50]. The formula for Cronbach’s α is shown in Equation (3), and the reliability interpretations are presented in Table 6.
(3)$$\alpha =\frac{k}{k-1}\left(1-\frac{\sum_{i=1}^{k}\sigma_{i}^{2}}{\sigma_{total}^{2}}\right)$$

Among them, *k* represents the number of items in the questionnaire, $\sigma_{i}^{2}$ denotes the variance of the ith item, and $\sigma_{total}^{2}$ represents the variance of the total scores of all items in the questionnaire.

Therefore, if Cronbach’s α is close to or above 0.70, the scale is considered to have acceptable reliability. If it is above 0.80, it is considered to have good reliability. An alpha value below 0.70 may indicate a need for further evaluation and potential improvement of the scale.

### 2.4. Logistic Regression Analysis

Logistic regression analysis is a statistical method used to explore the influence of independent variables (also known as predictor or explanatory variables) on a dependent variable (also known as the outcome or response variable), often involving binary outcomes (such as yes/no, success/failure) [51,52]. In this study, this method was employed to examine whether the attributes of the surveyed individuals significantly influence food waste. The defined model for the binary logistic regression probability function is as follows:(4)$$\ln\frac{P_{i}}{1-P_{i}}=Y=\beta_{0}+\beta_{1}\cdot X_{1}+\beta_{2}\cdot X_{2}+\ldots \ldots +\beta_{i}X_{i}+\varepsilon$$

Among these, *Y* represents the dependent variable, *X* represents the independent variable, *β_0_* is the intercept, *β*_1_*, β*_2_*, …, β_i_* are the regression coefficients, *Pi* indicates the probability that the dependent variable takes the value 1 given the independent variable *X_i_* (*i* = 1, 2, …, n), and (1 − *P_i_*) represents the probability that the dependent variable takes the value 0. The term *ε* represents the random error term. Food waste is treated as the dependent variable. In the coding of food waste occurrences, high frequency is represented as 1, corresponding to the response options “always” or “often” in the questionnaire; low frequency is represented as 0, corresponding to the response options “occasionally” or “never” in the questionnaire.

This study chose five individual characteristics as independent variables for analysis, including area (*X*_1_), gender (*X*_2_), age (*X*_3_), educational level (*X*_4_), and annual household income (*X*_5_). For detailed statistical details, please consult Table 7.

Through such analysis, we can understand which attributes of the respondents have significant impacts on food waste, providing a basis for formulating targeted measures to reduce food waste.

## 3. Results

### 3.1. Food Waste Carbon Footprint

#### 3.1.1. Amount of Food Waste Generated by Shanghai Residents

Shanghai is one of the most prosperous cities in China and also the first city in China to implement garbage sorting. In 2021, the total amount of domestic waste in Shanghai was 12.32 million tons [53]. From July 2019 to May 2022, compared to the first half of 2019 before the implementation of the “Shanghai Municipal Domestic Waste Management Regulations”, the separation of wet waste increased by 72.9% [54]. The survey data of this study is shown in Table 8. Cooking waste has the highest quantity, accounting for 52.53% of the total food waste. The next in line are leftover of cooked dishes, untouched food, leftover of cooked staple food, and tea leaves and coffee grounds, accounting for 18.84%, 15.67%, 8.91%, and 4.06%, respectively. The survey results indicate that the per capita daily food waste generated in Shanghai is 0.57 kg and the per capita annual food waste in Shanghai is 206.59 kg. Among them, 56.58% of the food waste is unavoidable, such as cooking waste and tea leaves and coffee grounds, while 43.42% of the food waste is edible and can be completely avoided or reduced. The results of the seventh population census show that the resident population of Shanghai is 24.87 million. Therefore, the daily food waste generated in Shanghai is 14,000 tonnes, and the annual food waste production amounts to 5,113,200 tonnes. In 2021, the annual grain production in Shanghai was 939,600 tonnes [53], and the leftover of cooked staple food accounted for 45.58 tonnes, which is equivalent to 48.51% of the grain production. Cooking waste refers to organic waste generated in the kitchen, such as food scraps, peels, vegetable leaves, and bones. It is, like tea leaves and coffee grounds, an unavoidable part of food waste. In the process of calculating the food waste carbon footprint in this study, we excluded cooking waste, tea leaves, and coffee grounds, and only calculated the carbon footprint of edible food waste. Edible food waste is entirely avoidable, meaning this portion of the carbon footprint can be completely eliminated.

#### 3.1.2. Carbon Footprint of Edible Food Waste by Residents in Shanghai

This article investigated the food waste situation of the interviewees and examined their activity data across various stages of the lifecycle, including the raw material production stage, transportation stage, food preparation stage, and waste disposal stage. After statistical analysis, the activity data for each stage are presented in Table 9 and Table 10. The annual edible food waste carbon footprint in Shanghai is 10,598,700 tCO_2_eq, and the per capita daily carbon footprint is 1.17 kgCO_2_eq. The total carbon emissions from food consumption by residents of Shanghai City in a year, including direct, indirect, and industrial carbon emissions, are approximately 3572.8 kgCO_2_eq. According to the calculations in this study, the annual carbon emissions from edible food waste produced by Shanghai residents amount to approximately 428.23 kgCO_2_eq, which accounts for 12% of the total carbon emissions from food consumption. Alternatively, if the reduction of edible food waste is achieved, Shanghai could reduce its annual CO_2_eq emissions by 10,598,700 tons.

In the process of food waste carbon footprint, the raw material production stage contributes the most, accounting for 48.13% of the overall footprint. The next highest contributor is the transportation stage, accounting for 39.79%. Therefore, implementing effective measures in the raw material production stage and transportation stage is key to reducing food waste. Additionally, the carbon emissions from the food preparation stage and food waste disposal stage account for 7.05% and 5.03%, respectively.

In the raw material production stage, the largest carbon footprint contribution comes from “Leftover of cooked dishes”, with a carbon footprint of 2.42 million tonnes of CO_2_eq, accounting for 47.52% of this stage. Next is “Untouched food”, with a carbon footprint of 2.25 million tonnes of CO_2_eq, accounting for 44.12% of the total carbon footprint in this stage.

In the transportation stage, the largest carbon footprint contribution comes from “Leftover of cooked dishes”, with a carbon footprint of 1.83 million tons of CO_2_eq, accounting for 43.39% of this stage. Next in line are “Untouched food” and “Leftover of cooked staple food, accounting for 36.09% and 20.52%, respectively.

In the food preparation stage, “Untouched food” do not require further processing, so its carbon footprint at this stage is 0. The carbon footprints of “Leftover of cooked staple food” and “Leftover of cooked dishes” are not high, accounting for 380,900 tCO_2_eq and 366,100 tCO_2_eq, respectively. The largest contributor to the carbon footprint at this stage is “Leftover of cooked staple food”.

In the food waste disposal stage, the carbon footprint is 532,800 tCO_2_eq, and the largest contributor is “Leftover of cooked dishes” with a carbon footprint of 231,200 tCO_2_eq, accounting for 43.39% of the total. Next in line are “Untouched food” and “Leftover of cooked staple food”, accounting for 36.09% and 20.52%, respectively.

### 3.2. The Factors Influencing Food Waste

#### 3.2.1. The Internal Consistency of the Questionnaire

This article used Cronbach’s α coefficient to assess the interrelatedness and overall consistency of the questionnaire, obtaining a value of 0.7855 (Table 11). For the current results, the Cronbach’s α coefficient is close to 0.8, indicating that the questionnaire exhibits a high level of internal consistency.

#### 3.2.2. The Impact of Respondents’ Characteristics on Food Waste

Being China’s economic and cultural nucleus, Shanghai has undergone swift urbanization, with an urbanization rate reaching 89.3% [25]. Additionally, Shanghai, as a destination for work, education, and lifestyle, has attracted a considerable number of young individuals, contributing to the relatively large proportion of the young population in the city. Table 12 presents a comprehensive overview of the profiles of the participants, encompassing aspects such as area, gender, age, educational level, and annual household income. Among the respondents, the urban population accounts for a substantial 91.54%, while the rural population comprises only 8.46%. This data highlights a prominent feature of Shanghai’s urbanization process. In terms of gender, the distribution is relatively balanced, with males accounting for 51.19% and females comprising 48.81% of the total respondents. The age group with the highest population is under 30 years old, accounting for 43.17%, followed by the age group of 31–40 years, which accounts for 37.75%. The highest number of respondents, accounting for 42.73%, are individuals with education from vocational or technical universities, totaling 197. Following this, those with a university education represent 33.19% of the sample. Additionally, there were no participants without formal education, indicating a favorable educational level in Shanghai. In terms of annual household income, the largest proportion is found within the range of CNY 150,001–300,000, accounting for 38.18% of the sample. A combined total of 84.87% of individuals have an annual household income exceeding CNY 150,000.

The attributes of the participants were examined utilizing a logistic regression model, and the findings of this analysis are presented in Table 13. In the assessment of significance, the odds ratio (OR) is employed to gauge the connection or impact between two categorical factors. This statistical metric is frequently utilized to evaluate the influence of an explanatory variable on the response variable [55,56]. When the odds ratio is equal to 1, it signifies absence of association between the two categories, meaning the probability of events occurring is the same for both groups. An odds ratio exceeding 1 denotes a positive connection between the independent and dependent variables. Conversely, an odds ratio of less than 1 signifies a negative association between the independent and dependent variables [29]. In addition, a hypothesis test for the odds ratio, known as the *p*-value, is also needed. If the null hypothesis (usually odds ratio equals 1, indicating no association) is true, the probability of obtaining results as extreme as or more extreme than the observed values is calculated. If the *p*-value is small (typically less than a predetermined significance level, such as 0.05), we reject the null hypothesis, indicating a significant association or effect of the odds ratio. If the *p*-value is large, we fail to reject the null hypothesis, and we cannot draw a conclusion of a significant association [57]. If the *p*-value (*p* > |z|) is below 0.05, it signifies statistical significance at the 95% confidence level. If it falls below 0.01, it implies statistical significance at the 99% confidence level.

The results show that the odds ratios (OR) of gender (*X*_2_), education level (*X*_4_), and annual household income (*X*_5_) are greater than 1, suggesting a more pronounced influence on the likelihood of food waste occurrence. On the other hand, the odds ratios of area (*X*_1_) and age (*X*_3_) are less than 1, implying a comparatively lesser effect on the likelihood of food waste occurrence. Incorporating the significance analysis results, despite the odds ratio (OR) value for gender (*X*_2_) being greater than 1, the *p*-value of 0.105 indicates that the impact of gender on the probability of food waste occurrence is not statistically significant. The *p*-value for education level (*X*_4_) is less than 0.05, indicating that education level has a significant impact on the probability of food waste occurrence. The *p*-value for annual household income (*X*_5_) is less than 0.01, indicating that annual household income has a highly significant impact on the probability of food waste occurrence. Additionally, both education level (*X*_4_) and annual household income (*X*_5_) show a positive correlation with food waste (*Y*), implying that higher education levels are associated with a higher frequency of food waste. Likewise, a rise in annual household income corresponds to a higher frequency of food waste. To sum up, education level and annual household income play pivotal roles in affecting food waste, whereas the influence of region, gender, and age on food waste occurrence might be comparatively minor.

#### 3.2.3. Factors Influencing the High Frequency of Food Waste Generation

In the survey, further investigation was conducted to understand the reasons behind the high frequency of food waste generation among respondents who reported highly frequent occurrences (i.e., often and always). The results are shown in Figure 3. According to the respondents’ answers, “excessive amount of foods (overabundance)” is the primary reason leading to a high frequency of food waste, accounting for 57.14% of the total. The second most prevalent reason is “passing the use-by date”, accounting for 42.86%. On the other hand, the least selected reason is “be bored and dissatisfied with it/them”, with less than 10% of the respondents choosing this option. Additionally, the other four reasons, including “having no plan to consume it/them shortly”, “already bought new ones”, “not delicious”, and “the deterioration of quality”, was chosen by respondents at a rate ranging from 20% to 30%.

Overall, these results indicate that food overabundance and exceeding the expiration date are the main reasons leading to a high frequency of food waste, while the impact of other factors is relatively small. To reduce food waste, it may be beneficial to strengthen consumers’ awareness of food purchasing and usage, encourage reasonable planning of food purchases, and enhance skills for food preservation and handling to avoid food expiration and waste. Additionally, promoting relevant policies and measures to encourage food businesses and the catering industry to take action in reducing food waste can also be effective. Conducting food waste education and awareness campaigns to increase public understanding and concern about food waste is also an essential step in addressing this issue.

### 3.3. Policies concerning Food Waste

This study specifically focused on the section related to food waste policies and utilized the Beijing University Legal Information Platform for policy retrieval. Relevant keywords, namely “food waste” and “grain conservation”, were set to ensure the screening of policy documents containing these keywords in their titles or full texts. From these policy documents, we filtered out the ones published at the national and Shanghai levels. This retrieval setting enabled us to accurately obtain policy files related to food waste issues and grain conservation, providing essential data sources and reference materials for our research. The timeline of major policies concerning food waste on both national and regional levels in Shanghai is shown in Figure 4.

Shanghai has consistently taken proactive measures to align with the national polices aimed at food waste reduction, and it can promptly take action and widely promote relevant measures. In 2010, the General Office of the State Council issued a notification on “Further strengthening the work of grain conservation and opposing food waste” [58]. In the same year, the Shanghai Municipal People’s Government also released a notification on “Further strengthening efforts in grain conservation and opposing food waste” [59]. Since then, extensive and in-depth initiatives have been carried out in Shanghai to promote food conservation and combat food waste. In 2013, the National Food and Strategic Reserves Administration issued guiding opinions on the “Grain Industry Taking the Lead in Loving Food, Saving Food, and Opposing Waste” [60]. In the same year, the Shanghai Grain Bureau issued a notice on “Further strengthening the work of promoting food conservation and opposing waste in the grain industry” [61]. Following that, notice of the “Opinions on Strengthening the Practice of Thriftiness and Opposing Food Waste” was issued by the General Office of the Central Committee of the Communist Party of China and the General Office of the State Council in 2014 [62]. In the same year, the Shanghai Municipal Grain Bureau issued a notice to the “National Grain Bureau to vigorously promote grain conservation, reducing losses, and opposing food waste” and other related documents [63].

Starting in 2020, China has experienced an official movement against food waste. China implemented its inaugural legislation addressing food waste, known as the “Anti-Food Waste Law of the People’s Republic of China”, on 29 April 2021 [64]. Afterward, Shanghai quickly took action and successively issued multiple policy documents in response to the national anti-food waste law. The latest anti-food waste policies in Shanghai were announced in March 2023. They include the “Notice on the Issuance of the Implementation Plan for Strengthening Food Waste Supervision in the Catering Sector” [65] and the “Notice on launching a special campaign to curb food waste in the catering industry” by the Shanghai Municipal Administration for Market Regulation [66]. However, the existing policies mainly target the catering industry, and there is still a lack of policies at the household or residential level.

## 4. Discussion and Recommendations

### 4.1. Discussion

The research results on the carbon footprint of food waste show that the raw material production stage has the highest carbon emissions, followed by the transportation stage, then the food waste preparation stage, and finally the food disposal stage. The view that the Raw Material Production Stage has the highest carbon emissions has gained some consensus among scholars [11,67,68]. Research on the carbon footprint of food waste has yielded differing perspectives. Some studies suggest that the transportation stage has a significant impact on the carbon footprint of food waste due to the substantial fuel consumption and carbon emissions involved in transporting food from production to consumption locations [69,70]. On the other hand, other studies take an opposing view, suggesting that the transportation stage accounts for a smaller proportion of the overall carbon footprint of food waste compared to other stages [11,38,68]. In this study, we align with the former perspective, which suggests that the transportation stage contributes significantly to the carbon footprint of food waste. The use of vehicles and energy during the transportation process releases a considerable amount of greenhouse gases, especially carbon dioxide. As the amount of food waste increases, so does the carbon emissions generated from its transportation. Therefore, when formulating strategies to reduce food waste, particularly considering the impact on carbon footprint, it is essential to prioritize environmental concerns at the transportation stage to minimize the adverse effects of climate change. The controversy surrounding the food waste treatment stage mainly revolves around two methods: composting and food waste biorefineries. Some scholars argue that composting is an effective way to reduce carbon emissions because it converts organic waste into fertilizer, reducing the amount of organic waste sent to landfills, and generating relatively low levels of carbon dioxide during the composting process. Therefore, they advocate for the promotion of composting as an environmentally friendly food waste treatment method [17]. On the other hand, other scholars believe that food waste biorefineries offer a more sustainable, environmentally friendly, and cost-effective approach. These biorefineries can convert organic materials from food waste into platform chemicals, biofuels, and other bio-based materials, providing sustainable resources for the production of various chemicals and materials. This method not only effectively manages food waste and reduces landfilling, but also alleviates environmental burdens and lowers carbon emissions, bringing both environmental and economic benefits to society [18]. In this study, we found that Shanghai currently adopts composting as the treatment method for wet food waste, which contributes the least to carbon emissions among the four stages. Therefore, it is an effective method for reducing carbon emissions. Food waste refinement may be a viable option worth considering, but the decision to implement food waste refinement needs to take into account various factors comprehensively.

Regarding the factors influencing household food waste, there are different viewpoints. The findings from this research suggest that education level and annual household income are two significant factors affecting food waste. Some studies suggest that higher education levels are associated with less food waste. This can be attributed to their enhanced discernment, as well as their reduced susceptibility to impulsive buying behavior [71,72]. Contrastingly, certain researchers maintain an opposing perspective. They believe that education level is positively correlated with the amount of food waste generated [73,74]. The findings of this research align with the latter view, indicating that higher education levels are associated with a higher frequency of food waste. Regarding annual household income, the majority of previous research has demonstrated a positive association between income and food waste [75,76], while an alternative viewpoint put forth by certain scholars suggests that household income might not be linked to food waste [77]. However, this view is not widely accepted. In this study, there is a positive correlation between annual household income and food waste. In summary, the factors influencing household food waste are complex and diverse, and different studies may yield different conclusions. Therefore, in addressing the issue of household food waste, it is essential to take into account various factors comprehensively.

In conclusion, each stage of the food waste carbon footprint contributes differently, and therefore, each stage should be given due attention. Measures need to be taken at various stages, including food production, transportation, preparation, and waste disposal, to reduce carbon emissions. Additionally, there is a need to enhance public awareness and educate consumers about the issue of food waste, encouraging responsible consumption and avoiding excessive purchasing and food wastage. Addressing the food waste carbon footprint is a comprehensive task that requires a holistic consideration of carbon emissions in each stage and the formulation of comprehensive emission reduction strategies. Through collective efforts and cooperation across society, we can effectively reduce carbon emissions from food waste and contribute to building a more sustainable food system.

### 4.2. Recommendation

1.Improving ecological agricultural production models.

Developing ecological agriculture is an essential means of carbon emissions reduction in the food production stage. Ecological agriculture emphasizes ecological balance, resource cycling, and the preservation of biodiversity, which contribute to reducing environmental burdens and thereby lowering carbon emissions. To promote ecological agriculture, the government can provide relevant policy support and incentives. Firstly, introducing policies that encourage the development of ecological agriculture to incentivize farmers to adopt ecological farming methods. Secondly, investing in research and promoting advanced technologies suitable for ecological agriculture, such as organic fertilizers, biopesticides, and water-saving irrigation systems, to enhance agricultural productivity and reduce carbon emissions. Additionally, encouraging farmers to adopt ecologically friendly planting and breeding methods to protect soil, water sources, and ecosystems. Simultaneously, strengthening international exchanges and cooperation in ecological agriculture, learning from the experiences of other countries, and advancing the global development of ecological agriculture. Through these measures, ecological agriculture can offer a more sustainable and environmentally friendly direction for the agricultural industry, achieving the dual goals of carbon emission reduction and ecological preservation.

2.Adopting a centralized supply and local consumption approach is recommended.

Shanghai is one of the largest cities in China, with a high degree of urbanization, leading to relatively limited land resources for agriculture. Despite the limited land resources, there are still some suitable areas for agricultural development, especially in the suburbs and surrounding regions. In these areas, agriculture can play a significant role, such as cultivating high-value agricultural products like vegetables, fruits, and flowers to meet the demand for fresh produce from urban residents. By promoting local agricultural production in Shanghai and implementing a centralized supply and local consumption approach, it is possible to minimize long-distance transportation, save energy, and reduce carbon emissions. This strategy not only facilitates the integration of urban and rural development but also contributes to the advancement of sustainable agriculture, making valuable contributions to ecological conservation and carbon reduction efforts.

3.Enhance waste sorting and resource utilization systems.

While Shanghai currently employs the effective strategy of utilizing wet waste resources, there remains an ongoing need for research and exploration into even more eco-friendly and sustainable methods to further reduce carbon emissions. Additionally, constructing advanced waste sorting and resource utilization facilities can help reduce energy consumption and carbon emissions during the waste processing process. Both government and society should actively encourage and support relevant research and innovations, raise public awareness of environmental protection, and promote waste sorting and resource recycling, all of which collectively contribute significantly to the sustainable development of Shanghai City. By continually refining waste management systems, encouraging innovation, and fostering environmental consciousness, the city can remain at the forefront of carbon emission reduction efforts while setting an example for others to follow.

4.Raise residents’ awareness of energy conservation.

Raising residents’ awareness of energy conservation has positive effects on carbon reduction and environmental protection, and it is a crucial aspect of achieving a low-carbon society and sustainable development. Conducting energy-saving awareness campaigns is a crucial step that can use various media formats to convey the importance of energy conservation and simple yet effective energy-saving methods to the public. In household life, rational planning of cooking processes to reduce unnecessary energy waste, such as choosing appropriate cookware and controlling flame size, as well as using water and electricity in moderation, are effective energy-saving measures.

5.Raise consumer awareness of food purchasing and utilization.

By conducting educational activities, promotional campaigns, and using media channels, raising public awareness about the impact of food waste on the environment and resources becomes an effective way to reduce food waste. Especially for individuals with higher education, emphasizing the significance of conserving food can stimulate their sense of responsibility and awareness in conserving food resources. Developing guidelines on food conservation, including providing tips on shopping, planning meals, and managing leftovers, can offer practical advice. Families can plan their shopping lists more effectively, avoid impulsive purchases, and buy food according to actual needs to reduce food leftovers. Additionally, enhancing residents’ economic awareness is crucial, encouraging prudent consumption and avoiding excessive purchasing and food waste. Through these measures, public environmental consciousness and the concept of food conservation can be raised, leading to reduced food waste at the household level and contributing to environmental protection and sustainable resource utilization.

6.Properly planning food storage.

“Excessive amounts of food (overabundance)” and “foods passing their use-by dates” are the primary reasons for high-frequency food waste. To reduce such waste, families should implement smart food storage planning. When storing food in the refrigerator or pantry, it is important to ensure proper ventilation and cooling to prevent food from rotting due to overcrowding. For frozen food, adequate packaging and sealing should be observed to prevent frost or spoilage. Timely handling of soon-to-expire food is also crucial. When shopping, pay attention to the expiration dates of food items and plan consumption accordingly. Prioritize the consumption of food items nearing their expiration dates to ensure they are fully utilized and reduce the likelihood of being discarded. Additionally, family members can coordinate their food consumption plans to avoid repetitive purchases and minimize waste. Understanding each family member’s food preferences and dietary habits is also essential for allocating food consumption time and proportions wisely to maximize utilization. By implementing these measures, families can effectively plan food storage and consumption, reduce food waste, and contribute to environmental protection and sustainable resource utilization.

7.Building a food donation platform.

Food donation platforms have been widely promoted and applied in many countries and regions. By promptly donating unsold but still edible food to those in need, these platforms further reduce carbon emissions from food waste. They create a crucial resource recovery channel in society, transforming soon-to-be expired food and surplus food from supermarkets and other establishments into valuable resources, benefiting individuals in need. The operation of food donation platforms usually involves food retailers, supermarkets, restaurants, and others donating food to charitable organizations, rescue groups, or community service centers, which then distribute the food to those who require assistance. Throughout this process, collaboration among various stakeholders can be fostered, creating a collective societal effort to address food waste. The widespread application and promotion of food donation platforms are essential for reducing carbon emissions from food waste, achieving a win-win situation for both environmental and social benefits.

## 5. Conclusions

This study took Shanghai as an example to estimate the carbon footprint of food waste and analyzed the influencing factors of consumer food waste. Additionally, the policies and strategies implemented by the country and Shanghai to combat food waste were reviewed. Based on these findings, recommendations were proposed to reduce food waste and its carbon emissions. The research findings indicate that: (1) The Shanghai municipal government attaches great importance to addressing the issue of food waste, responding promptly, and conducting extensive publicity, aligning with national policies. However, the current policies mainly target the catering industry, and there is a lack of specific policies addressing food waste at the household level. (2) In Shanghai, the per capita daily generation of food waste is 0.57 kg. Of this, 43.42% is edible food waste, which contributes to a per capita daily carbon footprint of 1.17 kgCO_2_eq. During the process of food waste carbon footprint generation, the food production and transportation stages are the highest contributors to carbon emissions. (3) The educational level and income level of consumers are the main factors influencing food waste generation. (4) Excessive food purchases and food expiration are also major reasons for high-frequency food waste. Based on the above research findings, the following recommendations are proposed: (1) Improve ecological agricultural production models. (2) Adopt a centralized supply and local consumption approach is recommended. (3) Enhance waste sorting and resource utilization system. (4) Raise residents’ awareness of energy conservation. (5) Raise consumer awareness of food purchasing and utilization. (6) Properly plan food storage. (7) Build a food donation platform. These recommendations offer feasible solutions to reduce food waste and its carbon emissions while promoting sustainable development in society. By enhancing public awareness of the issue of food waste, a collective effort from the entire society can be fostered to address this challenge. Future efforts should continue to focus on and improve these strategies, aiming to drive society toward a more environmentally friendly and sustainable direction.

There is also a portion of edible food waste within cooking waste. However, due to the lack of precise data regarding the proportion, this study excludes the portion of edible food waste from cooking waste when calculating the carbon emissions of edible food waste. As a result, the calculated carbon emissions in this study may be slightly lower than the actual amount produced. Posteriorly, given the constrained sample size, the data’s representativeness could be somewhat constrained as well, necessitating further data validation and more thorough analysis to enhance the credibility and generalizability of the research findings. Additionally, it is important to acknowledge that certain audience groups, such as specific social, age, or cultural backgrounds, may be unwilling or unable to participate in the survey. This absence of some audience groups could lead to a lack of representativeness in certain aspects of the survey results, potentially impacting a comprehensive understanding of the issue of food waste. To overcome these limitations, future research can adopt diverse data collection methods, enlarge the sample size, and attempt to cover a more extensive range of demographics. Moreover, employing techniques like in-depth interviews and focus groups may be beneficial in comprehending the various perspectives and needs of different audience groups, thereby gaining a more comprehensive understanding of the problem of food waste. In further research and policy making, we need to continue to pay attention to and address these limitations to ensure the scientific and practical value of the research findings.

## Figures and Tables

**Figure 1 foods-12-03211-f001:**
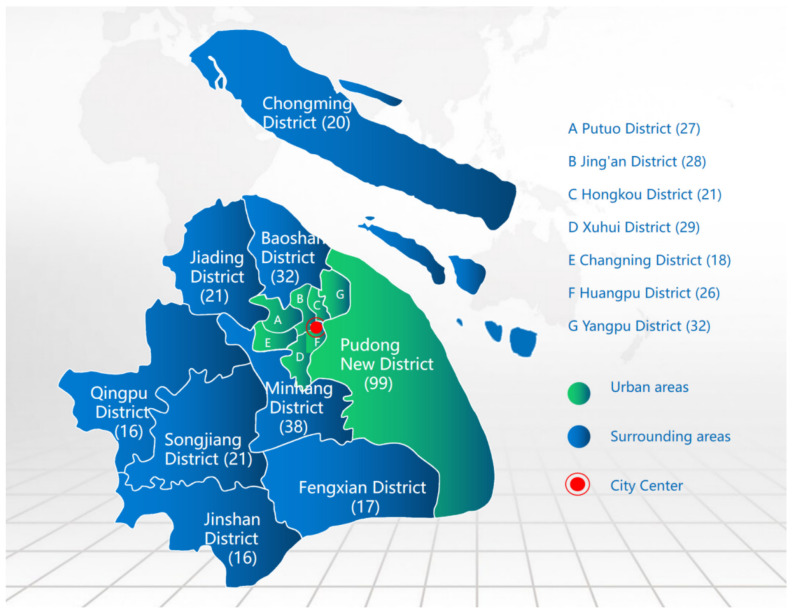
Sample size map of the public questionnaire.

**Figure 2 foods-12-03211-f002:**
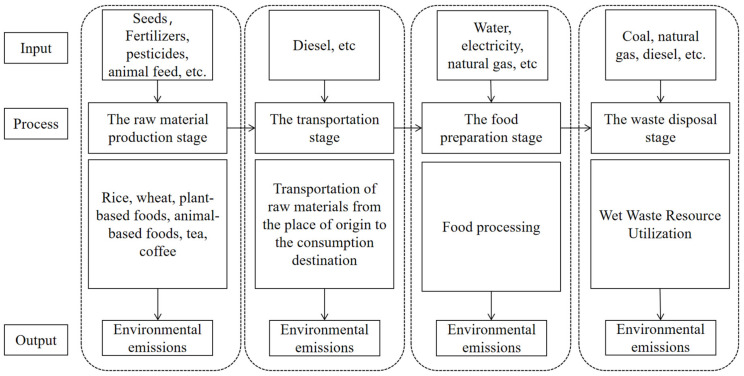
System boundary diagram for food waste.

**Figure 3 foods-12-03211-f003:**
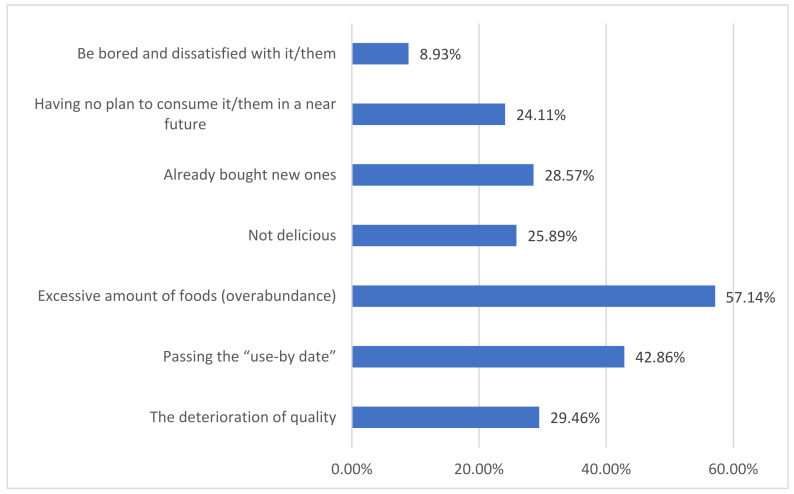
Factors influencing the high frequency of food waste generation.

**Figure 4 foods-12-03211-f004:**
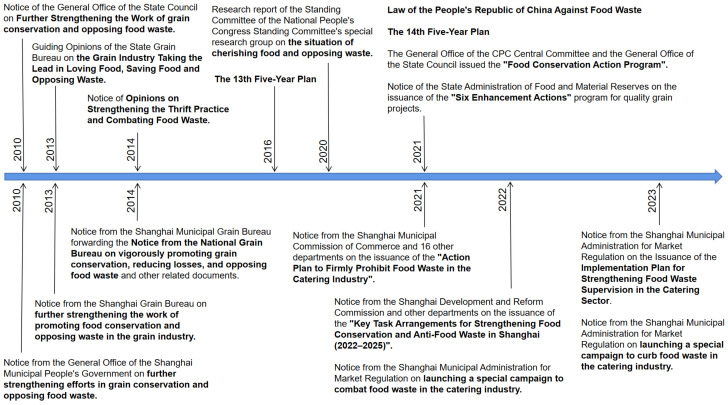
The timeline of major policies concerning food waste on both national and regional levels in Shanghai.

**Table 1 foods-12-03211-t001:** Carbon emission coefficients for the raw material production stage.

Project	Carbon Emission Coefficient	Unit
Leftover of cooked staple food	0.937	kgCO_2_eq/kg
Leftover of cooked dishes	2.516	kgCO_2_eq/kg
Untouched food	2.809	kgCO_2_eq/kg

**Table 2 foods-12-03211-t002:** Carbon emission coefficients for the transportation stage.

Project	Carbon Emission Coefficient	Unit
Diesel	1.9	kgCO_2_eq/kg

**Table 3 foods-12-03211-t003:** Carbon emission coefficients for energy in the food preparation stage.

Project	Carbon Emission Coefficient	Unit
Water	0.193	kgCO_2_eq/kg
Electricity	0.612	kgCO_2_eq/kg
Natural gas	1.881	kgCO_2_eq/kg

**Table 4 foods-12-03211-t004:** Carbon emission coefficients for the food preparation stage.

Project	Staple Food	Dishes
Carbon Emission Coefficient (kgCO_2_eq/kg)	0.836	0.38

**Table 5 foods-12-03211-t005:** Carbon emission coefficients for the food waste disposal stage.

Project	Carbon Emission Coefficient	Unit
Wet Waste Resource Utilization	0.24	kgCO_2_eq/kg

**Table 6 foods-12-03211-t006:** The interpretation of Cronbach’s α coefficient’s reliability.

Cronbach’s α Range	Reliability Interpretation
0.90 and above	Excellent reliability
0.80 to 0.89	Good reliability
0.70 to 0.79	Acceptable reliability
0.60 to 0.69	Questionable reliability
0.50 to 0.59	Poor reliability
Below 0.50	Very poor reliability

**Table 7 foods-12-03211-t007:** Selection and handling of model variables.

Variable Name	Definition and Allocation of Variables	Mean	Standard Deviation
Food waste (*Y*)	High frequency = 1,	0.24	0.43
	Low frequency = 0		
Area (*X*_1_)	Urban areas = 1,	0.92	0.28
	Surrounding areas = 0		
Gender (*X*_2_)	Male = 1,	0.51	0.50
	Female = 0		
Age (*X*_3_)	≤30 = 1,	1.79	0.82
	31–40 = 2,		
	41–50 = 3,		
	51–60 = 4,		
	>60 = 5		
Education level (*X*_4_)	No formal education = 1,	6.29	1.19
	Elementary school = 2,		
	Junior high school = 3,		
	Technical school = 4,		
	Senior high school = 5,		
	Vocational or Technical University = 6,		
	University = 7,		
	Master’s Degree or Higher = 8		
Annual household income (*X*_5_)	Up to CNY 30,000 = 1,	3.77	1.30
	CNY 30,001–80,000 = 2,		
	CNY 80,001–150,000 = 3,		
	CNY 150,001–300,000 = 4,		
	CNY 300,001–1 million = 5,		
	CNY 1–5 million = 6,		
	Over CNY 5 million = 7		

**Table 8 foods-12-03211-t008:** Food waste generation in residents’ daily life in Shanghai.

Project	Amount of Food Waste Generated per Capita per Day (g)	Annual per Capita Food Waste (kg)	Daily Food Waste Generated (10^4^t)	Annual Food Waste (10^4^t)
Cooking waste	297.30	108.51	0.74	268.57
Leftover of cooked staple food	50.43	18.41	0.12	45.56
Leftover of cooked dishes	106.63	38.92	0.26	96.33
Untouched food	88.67	32.37	0.22	80.11
Tea leaves and coffee grounds	22.97	8.39	0.06	20.75
Total	566.00	206.59	1.40	511.32

**Table 9 foods-12-03211-t009:** The annual carbon footprint of food waste in Shanghai’s residents’ daily lives.

Project	Annual Food Waste (10^4^t)	Carbon Footprint for the Raw Material Production Stage (10^4^tCO_2_eq)	Carbon Footprint for the Transportation Stage (10^4^tCO_2_eq)	Carbon Footprint for the Food Preparation Stage (10^4^tCO_2_eq)	Carbon Footprint for the Food Waste Disposal Stage (10^4^tCO_2_eq)
Leftover of cooked staple food	45.56	42.69	86.56	38.09	10.93
Leftover of cooked dishes	96.33	242.37	183.03	36.61	23.12
Untouched food	80.11	225.03	152.21	0	19.23
Total	222	510.09	421.8	74.7	53.28
The annual carbon footprint of food waste		1059.87 (10^4^tCO_2_eq)			

**Table 10 foods-12-03211-t010:** The carbon footprint of food waste in Shanghai.

Per Capita Daily Carbon Footprint of Food Waste (kgCO_2_eq)	Per Capita Annual Carbon Footprint of Food Waste (kgCO_2_eq)	Daily Carbon Footprint of Food Waste Generated (10^4^tCO_2_eq)	Annual Carbon Footprint of Food Waste (10^4^tCO_2_eq)
1.17	428.23	2.90	1059.87

**Table 11 foods-12-03211-t011:** The Cronbach’s α coefficient of the questionnaire.

Cronbach’s α	Number of Items
0.7855	7

**Table 12 foods-12-03211-t012:** Characteristics of participants.

Characteristics	Number of Respondents	Percentage
Area:		
Urban areas	422	91.54%
Rural areas	39	8.46%
Gender:		
Male	236	51.19%
Female	225	48.81%
Age:		
≤30	199	43.17%
31–40	173	37.53%
41–50	79	17.14%
51–60	8	1.74%
>60	2	0.43%
Education level		
No formal education	0	0
Elementary school	4	0.87%
Junior high school	21	4.56%
Technical school	12	2.60%
Senior high School	21	4.56%
Vocational or Technical University	197	42.73%
University	153	33.19%
Master’s Degree or Higher	53	11.50%
Annual household income		
Up to CNY 30,000	23	4.99%
CNY 30,001–80,000	62	13.45%
CNY 80,001–150,000	77	16.70%
CNY 150,001–300,000	176	38.18%
CNY 300,001–1 million	88	19.09%
CNY 1–5 million	27	5.86%
Over CNY 5 million	8	1.74%

**Table 13 foods-12-03211-t013:** The results of logistic regression model.

	Odds Ratio	Standard Error	z	*p* > |z|	(95% Conf. Interval)	
*X* _1_	0.715	0.282	−0.85	0.396	0.330	1.551
*X* _2_	1.443	0.326	1.62	0.105	0.926	2.248
*X* _3_	0.876	0.126	−0.92	0.360	0.661	1.162
*X* _4_	1.310	0.152	2.33	0.020 *	1.044	1.643
*X* _5_	1.375	0.128	3.42	0.001 **	1.146	1.650

* Indicates significance with a confidence level of 95%. ** Indicates significance with a confidence level of 99%.

## Data Availability

The data used to support the findings of this study can be made available by the corresponding author upon request.

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
