# Peer review of "Exploring Household Food Waste Reduction for Carbon Footprint Mitigation: A Case Study in Shanghai, China"

_foods, 2023, doi:10.3390/foods12173211_

Round 1
Reviewer 1 Report
The manuscript title is poorly constructed. ‘As one of China's major cities and a key global economic and cultural hub, Shanghai's food waste carbon emissions hold considerable significance. The main focus of this study is to estimate the generation of household food waste and the carbon footprint of edible food waste in Shanghai.’ – say directly, avoid repetitive content. Replace it/they with the proper words to avoid confusion. GHG must be placed next to the full phrase (greenhouse gas) first. ‘According to the International Energy Agency (IEA), it is estimated that in 2022, global energy-related carbon dioxide (CO2) emissions grew by 321 mt’ – we are in 2023, specify 2022 data, not estimates. Check for no or extra spaces throughout the manuscript. E.g., ‘tonnes.It’. A lot of sentences are too short. E.g., these four sentences can be easily reconfigured as one: ‘China is one of the top GHG emitters. It accounts for 27.79% of global emissions. In 2018, it emitted 13,739.79 million tonnes.It has had one of the biggest increases 250% in GHG emissions since 1990’. A lot of sentences include zero info. E.g., ‘However, the impact of food waste on the environment is increasingly severe.’ A lot of data are too old to reflect the current picture or are too particular for generalization. Why using alternatively % and percent? Many times you use too many (often repetitive) words to express an idea. E.g., ‘A study assessed the carbon emissions from food waste in Korean households. The study found that food waste’. A lot of content is not substantiated. E.g., ‘In recent years, people have gradually realized that food waste not only leads to economic losses and resource wastage but also has significant environmental impacts. As a result, an increasing number of scholars have started conducting in-depth research on the carbon footprint of food waste. Household food waste occupies a significant position in the entire food supply chain and makes a substantial contribution to the total amount of food waste. Every household generates a certain degree of food waste, making it a widespread issue overall.’ What is the point of adding Chinese characters on pp. 14–15? ‘The data used in this study is reliable and authentic’ – what is the point of including this? The reference list is extremely poorly edited, a lot of sources are old and the proportion of non-peer reviewed content is high.
The relationship between environmental costs and organic management as regards carbon footprint impact of household food waste has not been covered, and thus such sources can be cited:
Lăzăroiu, G., Andronie, M., Uţă, C., and Hurloiu, I. (2019). “Trust Management in Organic Agriculture: Sustainable Consumption Behavior, Environmentally Conscious Purchase Intention, and Healthy Food Choices,” Frontiers in Public Health 7: 340. doi: 10.3389/fpubh.2019.00340.
Pocol, C.B., Amuza, A., Moldovan, M.G., Stanca, L., Dabija, D.C. 2023. Clustering food wasters on an emerging market: a national wide representative research. Foods, 12(10), 1973. https://doi.org/10.3390/foods12101973
Lăzăroiu, G., Valaskova, K., Nica, E., Durana, P., Kral, P., Bartoš, P., et al. (2020). “Techno-Economic Assessment: Food Emulsion Waste Management,” Energies 13(18): 4922. doi: 10.3390/en13184922.
The manuscript title is poorly constructed. ‘As one of China's major cities and a key global economic and cultural hub, Shanghai's food waste carbon emissions hold considerable significance. The main focus of this study is to estimate the generation of household food waste and the carbon footprint of edible food waste in Shanghai.’ – say directly, avoid repetitive content. Many times you use too many (often repetitive) words to express an idea. E.g., ‘A study assessed the carbon emissions from food waste in Korean households. The study found that food waste’.
Reviewer 2 Report
The paper introduces ideas that might stimulate others to approach the complex domain of food waste. Directions could be reducing its amount or/and, if that is not possible, designing proper methodologies to include them in the circular economy. The food waste sector has a significant impact on the environment. The authors could consider the following observations to improve the article's consistency.
· Use subscript when the situation requests it (e.g., chemical formula);
· The manuscript refers to the use of a questionnaire. You may examine the possibility of including the socio-demographic characteristics of the respondents. It is only mentioned that they were over 18 years old. Data such as gender, level of education and income, and age interval could be correlated to their responses and determine the proper directions to reduce the carbon footprint determined by this sector. You mentioned in the Abstract that ”Education level and annual household income significantly influence food waste generation, and excessive food quantities and expiration dates lead to high-frequency food waste”. The affirmation might be sustained by including the information suggested. There could also be correspondence with the reports available in the literature and underline the differences and similarities found.
· Specify the type of the question utilized (e.g., scaled, semi-open, closed, factual, etc.);
· The intention of the investigation had to be based on pre-existing hypotheses based on which the survey queries were made. Please highlight them and specify if the respondent's answers confirmed these.
The paper could be considered for publication after major changes. It has to be revised by the authors and resubmitted with suggested modifications specified in the reviewer’s comments.
Review the English style. In some sections, there is excessive use of the same word. Try to use synonyms or rephrase. (e.g.,” survey”);
Round 2
Reviewer 1 Report
This revised version can be published.
Reviewer 2 Report
The authors responded only to some of the recommendations previously made.
In the material uploaded there are no included any concrete data regarding the respondents socio demographic information (e.g., a sumarising table of these). The conclusions regarding the hypothesys formulated are not sustaind through statistic tests that could confirm them.
Minor editing of English language required.
